Design of new acid-activated cell-penetrating peptides for tumor drug delivery

Yao Jia 1
Ma Yinyun 2
Zhang Wei 3
Li Li 2
Zhang Yun 2
Zhang Li 2
Liu Hui 2
Ni Jingman nijm@lzu.edu.cn 2
Wang Rui 3
1 The First Hospital, Lanzhou University , Lan Zhou , Gansu Province , China
2 School of Pharmacy, Lanzhou University , Lan Zhou , Gansu Province , China
3 Key Laboratory of Preclinical Study for New Drugs of Gansu Province, Lanzhou University , Lan Zhou , Gansu Province , China
Diao Jiajie
Electronic publication date: 2017 Jun 7
Publication date: 2017
Volume: 5
Electronic Location ID: e3429
Received 2017 Jan 11; Accepted 2017 May 17
Copyright: ©2017 Yao et al.
Copyright year: 2017
Copyright holder: Yao et al.
License: This is an open access article distributed under the terms of the Creative Commons Attribution License, which permits unrestricted use, distribution, reproduction and adaptation in any medium and for any purpose provided that it is properly attributed. For attribution, the original author(s), title, publication source (PeerJ) and either DOI or URL of the article must be cited.
License URL: https://creativecommons.org/licenses/by/4.0/

Keywords: Cell-penetrating peptide, PH-sensitivity, TH analogs, Antitumor

Funding: National Natural Science Foundation of China 81473095 81273440 91213302 Ministry of Science and Technology of China 2012ZX09504-001-003 Doctoral Program of Higher Education 20130211130005 Central Universities lzujbky-2013-151 lzujbky-2015-308 lzujbky-2016-144 Natural Science Foundation of Gansu Province 145RJZA075 This work was supported by the National Natural Science Foundation of China (Nos. 81473095, 81273440 and 91213302), the Key National S&T Program “Major New Drug Development” of the Ministry of Science and Technology of China (No. 2012ZX09504-001-003), Specialized Research Fund for the Doctoral Program of Higher Education (No. 20130211130005), Research Funds for the Central Universities (lzujbky-2013-151, lzujbky-2015-308, lzujbky-2016-144) and Natural Science Foundation of Gansu Province (145RJZA075). The funders had no role in study design, data collection and analysis, decision to publish, or preparation of the manuscript.

==============================
TH(AGYLLGHINLHHLAHL(Aib)HHIL-NH2), a histidine-rich, cell-penetrating peptide with acid-activated pH response, designed and synthesized by our group, can effectively target tumor tissues with an acidic extracellular environment. Since the protonating effect of histidine plays a critical role in the acid-activated, cell-penetrating ability of TH, we designed a series of new histidine substituents by introducing electron donating groups (Ethyl, Isopropyl, Butyl) to the C-2 position of histidine. This resulted in an enhanced pH-response and improved the application of TH in tumor-targeted delivery systems. The substituents were further utilized to form the corresponding TH analogs (Ethyl-TH, Isopropyl-TH and Butyl-TH), making them easier to protonate for positive charge in acidic tumor microenvironments. The pH-dependent cellular uptake efficiencies of new TH analogs were further evaluated using flow cytometry and confocal laser scanning microscopy, demonstrating that ethyl-TH and butyl-TH had an optimal pH-response in an acidic environment. Importantly, the new TH analogs exhibited relatively lower toxicity than TH. In addition, these new TH analogs were linked to the antitumor drug camptothecin (CPT), while butyl-TH modified conjugate presented a remarkably stronger pH-dependent cytotoxicity to cancer cells than TH and the other conjugates. In short, our work opens a new avenue for the development of improved acid-activated, cell-penetrating peptides as efficient anticancer drug delivery vectors.

Introduction

Cancer is a major socio-economic burden on the society. Chemotherapy is an effective conventional treatment against cancer, however, traditional chemotherapy drawbacks such as limited selectivity, deleterious side effects, and multi-drug resistance, seriously restrict their clinical curative effects (Chari, 2008; Monsuez et al., 2010; Monje & Dietrich, 2011). Therefore, the development of antitumor drugs with high specificity and diminished side effects remains a considerable challenge.

Cell-penetrating peptides (CPPs) are typically short cationic sequences of 10–30 amino acids. Due to the ability to traverse cell membranes, CPPs are widely used in the cellular delivery of proteins (Ana, Wei-Ming & Chin, 2016), plasmid DNA (Kato et al., 2016), oligonucleotides (Helmfors, Eriksson & Langel, 2015), and liposomes (Huang et al., 2013). Meanwhile, due to their aqueous solubility, and tissue penetration and distribution capabilities, CPPs are also increasingly used for delivery of anticancer drugs (Vivès, Schmidt & Pèlegrin, 2008; Fonseca, Pereira & Kelley, 2009). Importantly, CPPs have the potential to overcome multidrug resistance (Davoudi et al., 2014; Vargas et al., 2014). When CPPs were coupled to traditional anti-tumor drugs such as paclitaxel (PTX) and podophyllotoxin (PPT), they significantly enhanced drug sensitivity of resistant cancer cell lines (Dubikovskaya et al., 2008; Lindgren et al., 2006). However, their major limitations in delivery applications, including non-specificity, severe toxicity due to their positive charge, and rapid clearance from blood, prevented their full clinical utilization.

In recent years, pH-responsive CPPs have been developed for targeted delivery based on the pH gradient between the tumor tissues and physiological environment (Lee, Gao & Bae, 2008; Shi et al., 2015; Jiang et al., 2012; Fei et al., 2014). Due to excess lactic acid secreted by solid tumors, the tumor extracellular environment shows a lower pH in comparison to normal physiological conditions (Tannockand & Rotin, 1989; Jähde, Rajewsky & Baumgärtl, 1982; Cardone, Casavola & Reshkin, 2005). The acidic tumor microenvironment mandates the development of pH-responsive CPPs for improved selectivity of anti-cancer drugs (Liang et al., 2014; Han et al., 2015). The initial electrostatic interaction between anionic cellular membrane surfaces and cationic CPPs is believed to play a major role in their cellular uptake (Ziegler, 2008; Guo, Peng & Kang, 2016; Futaki et al., 2007). Thus, modulating this electrostatic interaction by masking or restoring the positive charge can significantly enhance the specificity of CPPs. Histidine, a unique amino acid, has no charge under physiological conditions, whereas it becomes protonated in acidic tumor microenvironments and exhibits pH-dependent cellular uptake with a high affinity for cell membranes (Zaro, Fei & Shen, 2012). A novel acid-activated CPP, TH, was designed and synthesized by our group by replacing all lysine residues of TK (AGYLLGKINLKKLAKL(Aib)KKIL-NH2) with histidine residues, which exhibited the desired features. At the same time, the toxicity of TH decreased under both in vitro and in vivo conditions compared to TK (Zhang et al., 2011).

However, due to the imidazole on histidine with a pKa around 6.5, histidine could protonate into a net positive charge at pH values below 6.5 (Zaro, Fei & Shen, 2012; Ouahab et al., 2014), thereby activating the membrane penetrating activity of TH and allowing its entry into the cells. However, histidine becomes relatively insensitive to the tumor extracellular environment when the pH is in the 6.5–7.2 range, thereby reducing the pH-dependent penetrating activity of TH. In order to extend the applications of acid-activated CPPs, it is necessary to improve their pH-response. The ability of the imidazole group to accept protons is a key factor for protonation. Electron donating groups could render the N atom on the imidazole ring more alkaline and contribute to its ability to receive a proton, consequently facilitating the increase in pKa value.

In order to facilitate easier protonation of acid-activated CPPs and render them more sensitive to a weakly acidic environment, we introduced alkylated histidine analogues to acid-activated TH, for a more sensitive, acid-targeted drug delivery. Electron donating groups (methyl, ethyl, isopropyl, and butyl) were introduced to the imidazole group of L-histidineto construct a series of histidine analogs, L-histidine(methyl), L-histidine(ethyl), L-histidine(isopropyl) and L-histidine(butyl). These histidine analogs were further introduced to TH, forming methyl-TH, ethyl-TH, isopropyl-TH, and butyl-TH analogs. The pH-dependent cellular uptake efficiencies of the new TH analogs were evaluated by flow cytometry and confocal laser scanning microscopy (CLSM), respectively. Importantly, the toxicities of these new TH analogs were examined. The antitumor drug CPT was conjugated to the new TH analogs, and its pH-dependent cytotoxicity to cancer cells was compared. In short, this work explores a new avenue for development of improved, acid-activated CPPs as potential anticancer drug vectors for efficient targeted drug delivery.

Materials and Methods

Materials

Rink amide MBHA resin, protected amino acids, and other reagents and solvents for peptide synthesis were purchased from GL Biochem Ltd (Shanghai, China). Trifluoromethanesulfonic anhydride was purchased from Energy Chemical (Shanghai, China). Trifluoroacetic acid (TFA), DMF, piperidine, methanol and dichloromethane were of analytical grade and distilled before use. All other reagents and solvents were obtained from Tianjin Reagent Chemical Co. 3-(4,5-Dimethylthiazol-2-yl)-2,5-diphenyltetrazolium bromide (MTT) was purchased from Sigma-Aldrich (St. Louis, MO, USA), while fluorescein isothiocyanate (FITC) was purchased from Aladdin Reagent Co. Ltd. (Shanghai, China). LDH Cytotoxicity Assay Kit was purchased from Promega Co. Ltd. (Beijing, China).

Synthesis of histidine analogs

Synthesis of L-his(R)-carbomethoxy

(1)L-his(methyl)-carbomethoxy trifluoromethanesulfonic anhydridem (0.632 ml, 3.75 mmol) was dissolved in anhydrous dichloromethane (10 ml), and stirred under argon atmosphere for 30 min at −75 °C. Then, methanol (3.75 mmol) and DIEA (0.65 ml, 3.75 mmol) in anhydrous dichloromethane (10 ml) were added dropwise into the solution. The solution was stirred for 30 min at −75 °C, followed by the addition of L-histidine methyl ester (2.5 mmol) in anhydrous dichloromethane (10 ml) to the reaction flask. The mixture was then allowed to gradually warm to room temperature with overnight stirring (Qian, Liu & Burke, 2011). After confirming the formation of L-histidine-carbomethoxy by thin layer chromatography, the solution was washed with saturated sodium bicarbonate solution twice, and saturated sodium chloride solution two more times. Solvents were removed by rotary evaporation to obtain dried (Na2SO4). The oily solid (5 mmol) was dissolved in dichloromethane. Then, trifluoroacetic acid (3.888 ml, 50 mmol) and triisopropylsilane (1.1 ml, 5.5 mmol) were added into the round-bottom flask containing the solid, and stirred for 4 h at room temperature. The mixture was concentrated to a gum and purified by silica gel flash chromatography (CH2Cl2:MeOH, from 40:1 to 10:1), with a yield of 80%.

(2)L-his(ethyl)-carbomethoxy trifluoromethanesulfonic anhydride (0.632 ml, 3.75 mmol) was dissolved in anhydrous dichloromethane (10 ml), and stirred under argon atmosphere for 30 min at −75 °C. Then, ethanol (3.75 mmol) and DIEA (0.65 ml, 3.75 mmol) were dissolved in anhydrous dichloromethane (10 ml) and added dropwise to the solution. After stirring for 30 min at −75 °C, L-histidine methyl ester (2.5 mmol) was dissolved in anhydrous dichloromethane (10 ml) and added to the reaction flask. After stirring for 3 h at −75 °C, the temperature was raised to −40 °C and stirred overnight, followed by stirring at room temperature for 8 h. The formation of L-histidine-carbomethoxy was confirmed by thin layer chromatography, after which the solution was washed with saturated sodium bicarbonate twice, and saturated sodium chloride two more times. Solvents were removed by rotary evaporation to provide dried (Na2SO4) mixture as a gel. The oily solid (5 mmol) was dissolved in dichloromethane; then, trifluoroacetic acid (3.888 ml,50 mmol) and triisopropylsilane (1.1ml, 5.5 mmol) were added into the round-bottom flask containing the solid, and stirred for 4 h at room temperature. The mixture was concentrated to a gum and purified by silica gel flash chromatography (CH2Cl2:MeOH, from 40:1 to 10:1), with a yield of 70%.

(3)L-his(isopropyl)-carbomethoxy trifluoromethanesulfonic anhydride (0.632 ml, 3.75 mmol) was dissolved in anhydrous dichloromethane (10 ml), and stirred under argon atmosphere for 30 min at −75 °C. Then, isopropanol (3.75 mmol) was dissolved in anhydrous dichloromethane (10 ml) with DIEA (0.65 ml, 3.75 mmol) and added dropwise to the solution. After stirring for 30 min at −75 °C, L-histidine methyl ester (2.5 mmol) was dissolved in anhydrous dichloromethane (10 ml) and added to the reaction flask and stirred for 4 h at −75 °C. Then, the temperature was raised to −40 °C and the solution was stirred for 3 h. Finally, the mixture was allowed to gradually warm to room temperature with overnight stirring. The formation of L-histidine-carbomethoxy was confirmed by thin layer chromatography, after which the solution was washed with saturated sodium bicarbonate solution twice, and saturated sodium chloride solution two more times. Solvents were removed by rotary evaporation to provide dried (Na2SO4) mixture as a gel. The oily solid (5 mmol) was dissolved in dichloromethane. Then, trifluoroacetic acid (3.888 ml,50 mmol) and triisopropylsilane (1.1 ml, 5.5 mmol) were added into the round-bottom flask containing the solid, and stirred for 4 h at room temperature. The mixture was concentrated to a gum and purified by silica gel flash chromatography (CH2Cl2:MeOH, from 90:1 to 40:1), with a yield of 38.7%.

(4)L-his(butyl)-carbomethoxy trifluoromethanesulfonic anhydride (0.632 ml, 3.75 mmol) was dissolved in anhydrous dichloromethane (10 ml), and stirred under argon atmosphere for 30 min at −75 °C. Then, butanol (3.75 mmol) was dissolved in anhydrous dichloromethane (10 ml) with DIEA (0.65 ml, 3.75 mmol) and added dropwise to the solution. After stirring for 30 min at −75 °C, L-histidine methyl ester (2.5 mmol) was dissolved in anhydrous dichloromethane (10 ml) and added to the reaction flask, with stirring for 4 h at −75 °C. Then, the temperature was raised to −40 °C and the mixture was stirred for 3 h, after which it was allowed to gradually warm to room temperature with overnight stirring. After the formation of of L-histidine-carbomethoxy was confirmed by thin layer chromatography, the solution was washed with saturated sodium bicarbonate solution twice, followed by saturated sodium chloride solution for two more times. Solvents were removed by rotary evaporation to provide dried (Na2SO4) mixture as a gel. The oily solid (5 mmol) was dissolved in dichloromethane. Then, trifluoroacetic acid (3.888 ml, 50 mmol) and triisopropylsilane (1.1 ml, 5.5 mmol) were added to the round-bottom flask containing the solid, and stirred for 4 h at room temperature. The mixture were concentrated to a gum and purified by silica gel flash chromatography (CH2Cl2:MeOH, from 80:1 to 40:1), with a yield of 36.4%.

Synthesis of L-histidine-carbomethoxy

Fmoc-histidine(Trt)-OH (3.1 g, 5 mmol) was dissolved in anhydrous after which thionyl chloride (1.49 ml, 20 mmol) was added dropwise to the solution with stirring in an ice bath. Then, the solution was stirred for 2 h at room temperature. After thin layer chromatography confirmed the disappearance of histidine, solvents were removed by rotary evaporation. White precipitate was dissolved in dichloromethane and washed with a saturated solution of sodium carbonate twice. Solvents were removed by rotary evaporation to provide dried (Na2SO4), at 92% yield.

Synthesis of L-his(R)-carbomethoxy

Trifluoromethanesulfonicanhydride (0.632 ml, 3.75 mmol) was dissolved in anhydrous dichloromethane (10 ml), and stirred under argon atmosphere at −75 °C for 30 min. Then, alcohols (such as ethanol, isopropanol, butanol, 3.75 mmol) and DIEA (0.65 ml, 3.75 mmol) were dissolved in anhydrous dichloromethane (10 ml) and added dropwise to the solution. After stirring for 30 min at −75 °C, L-histidine methyl ester (2.5 mmol) was dissolved in anhydrous dichloromethane (10 ml) and added to the reaction flask, with stirring for 4 h at −75 °C. The mixture was then allowed to gradually warm to room temperature with overnight stirring (Qian, Liu & Burke, 2011). The formation of L-histidine-carbomethoxy was confirmed by thin layer chromatography, and the solution was washed with saturated sodium bicarbonate twice, and saturated sodium chloride solution for two more times. Solvents were removed by rotary evaporation to provide dried (Na2SO4) mixture as a gel. The oily solid (5 mmol) was dissolved in dichloromethane. Then, trifluoroacetic acid (3.888 ml, 50 mmol) and triisopropylsilane (1.1 ml, 5.5 mmol) were added into the round-bottom flask containing the solid, and stirred for 4 h at room temperature. The mixture was concentrated to a gum and purified by silica gel flash chromatography (CH2Cl2:MeOH, from 80:1 to 40:1), with a yield of 38.7%.

Synthesis of L-his(R)

Compound 2 was dissolved in THF, and 4.8% LiOH⋅H2O was added dropwise to the reaction flask in an ice bath and stirred for 5 min. Then, the solution was acidified to pH 5 using aqueous HCl (1 mmol), after which ethyl acetate was added to the solution to extract four times. The organic layer was dried using anhydrous sodium sulfate concentrated in vacuo. Product 3 was obtained and the structure was detected by 1H-NMR(300 MHz, CDCl3).

Synthesis of TH analog peptides

All peptides shown in Table 1 were synthesized on a MBHA (0.43mmol/g) resin using the standard Fmoc-chemistry-based strategy. Fluorescein moiety (FITC) was attached to the N-terminusvia an aminohexanoic acid spacer by treating the resin-bound peptide (0.1 mmol) with FITC (0.1 mmol) and diisopropyl ethyl amine (0.5 mmol) in DMF for 12 h. All crude peptides were purified and analyzed by reversed-phase high performance liquid chromatography (RP-HPLC) on a C18 column, and then characterized by electrospray ionization mass spectrometry (ESI-MS).

Table 1 Amino acid sequence, physico-chemical properties of TH and its analogs.

Peptides	Amino acid sequences	Molecular mass	RP-HPLC retention times (min)b	
		Calculated	Measureda		
TH	AGYLLGHINLHHLAHL(Aib)HHIL-NH2	2,365	2365.39	18.539	
Methyl-TH	AGYLLGHMINLHMHMLAHML(Aib)HMHMIL-NH2	2,448	2448.44	18.058	
Ethyl-TH	AGYLLGHEINLHEHELAHEL(Aib)HEHEIL-NH2	2,532	2532.53	18.623	
Isopropyl-TH	AGYLLGHIINLHIHILAHIL(Aib)HIHIIL-NH2	2,616	2616.63	19.041	
Butyl-TH	AGYLLGHBINLHBHBLAHBL(Aib)HBHBIL-NH2	2,700	2700.74	20.561	
Notes.

a Molecular masses were determined by electrospray ionization mass spectrometry.

b RP-HPLC retention time was measured by analytical HPLC with sunfire TM column (3.9 mm × 150 mm, Waters).

HM L-his(Methyl)

HE L-his(Ethyl)

HI L-his(Isopropyl)

HB L-his(Butyl)

All peptides were synthesized with cysteine on the N-terminus, with the thiol group serving as the attachment site for the cleavable linker. Following a procedure reported in previous literature (Henne, Doorneweerd & Hilgenbrink, 2006), CPT was attached to the cysteine of peptides with a disulfide carbonate releasable linker.

Cell culture

Hela cells used in this study were obtained from Laboratory Center for Medical Science of Lanzhou University. Cells were maintained in RPMI1640 medium (Gibco BRL,Gaithersburg, MD, USA) containing 10% heat-inactivated neonatalbovine serum (NBS) (Sijiqing Biotech, Hangzhou, China), penicillin (100 IU/mL) and streptomycin (100 µg/mL). All cell lines were cultured in a 5% CO2 humidifiedatmosphere at 37°C.

In vitro cellular uptake study

Quantitative cellular uptake assay

To access whether TH analogs exhibited better pH-sensitivity than TH, HeLa cells (1 × 105 cells/well) were plated into 24 well microplates and cultured for 24 h. After one hour long incubation with serum-free medium containing 5 µM FITC- labeled peptides and adjusted to pH 7.4, 6.5 or 6.0, the cells were washed twice with PBS and then incubated with 0.02% trypsin for one min. The cells were harvested and centrifuged at 1,000 rpm for 5 min. Subsequently, the cell pellet was suspended and washed twice with PBS and resuspended in PBS. Finally, fluorescence intensity of 10,000 cells was analyzed with an FACS caliber flow cytometer using 488-nm laser excitation.

Confocal laser scanning microscopy

To obtain a direct insight into the distribution of FITC-labeled peptides in cells, HeLa cells (6 × 104 cells/well) were plated in a glass-bottom culture dish for 24 h and then incubated with the FITC-labeled peptides using conditions and concentrations described above. After 1 h incubation at 37°C, the cells were washed with cold PBS three times and imaged using confocal laser scanning microscopy.

In vitro cytotoxicity assays

MTT assay

Cytotoxicity of all peptides against HeLa cells was evaluated using the MTT assay under varying pH conditions (pH 7.4, 6.5 and 6.0). Cells were seeded at 1 × 104 cells/well in 96-well plates, 24 h before treatment. After washing, cells were treated with serum-free medium, adjusted to pH 7.4, 6.5 or 6.0, and containing various concentrations of peptides. After 2 h incubation, 10 µL of MTT (5 mg/mL) was added to each well and incubated for another 4 h at 37°C. Then, the medium was removed and cells were dissolved by 150 µL dimethyl sulfoxide. Absorbance was determined using a microplate reader at 490 nm. Cell viability (%) was calculated using the following equation: Atest∕Acontrol × 100%, where Atest and Acontrol represent the absorbance of cells treated with different test solutions and blank culture media, respectively.

Lactate dehydrogenase (LDH) leakage assay

Membrane integrity was evaluated using the CytoTox-ONE assay. Cells were seeded at 1 ×104cells/well in a 96-well plate 24 h before treatment. After 2 h incubation with 100 µL serum-free medium, adjusted to pH 7.4, 6.5 or 6.0 and containing various concentrations of peptides, 40 µL of medium was transferred to a black fluorescence plate and incubated for 10 min with 40 µL of CytoTox-ONE reagent, followed by 20 µL of stop solution. Fluorescence was recorded with an excitation wavelength of 560 nm and emission wavelength of 590 nm. Untreated cells were defined as those where no leakage was observed while 100% leakage was defined as a total release of LDH due to cell lysis in 0.2% Triton X-100.

Hemolysis assays

To evaluate the safety of the peptide for application, hemolysis assays were performed. Fresh mouse blood was collected in heparin sodium-containing centrifuge tubes. Erythrocytes were separated by centrifugation at 800×g for 10 min and washed three times with phosphate buffered saline (PBS). Obtained erythrocytes were resuspended in PBS to 8 % (v/v). 100 µL of the peptide solution of various concentrations and 100 µL of the erythrocyte suspension were added to the wells of a 96-well plate. PBS and 0.2 % TritonX-100 were used as agents for 0 % and 100 % hemolysis, respectively. Plates were incubated for 1 h at 37°C and centrifuged at 1,200×g for 15 min. A total of 100 µL of each supernatant was transferred to a 96-well plate, and the release of hemoglobin was determined using a microplate reader at 490 nm.

In vitro antitumor efficacy

The antiproliferative effects of all conjugates were determined by the MTT assay. Cells were seeded in 96-well plates at a density of 5 × 103 cells/well. After 30 min incubation with serum-free medium adjusted to pH 7.4, 6.5 or 6.0 and containing various concentrations of agents, the remaining agents that did not enter the cells were washed to remove the old medium. After 72 h, the cytotoxicity of relevant agents was determined by the MTT assay as described earlier.

Results

Design and synthesis of acid-activated TH analogs

The detailed synthesis protocol of L-his(R) is shown in Fig. 1. Fmoc-histidine(Trt)-OH reacted with methanol to form L-histidine-carbomethoxy, thereby protecting the carboxyl of L-histidine. Then, electrophilic substitution reaction was used to introduce electron donating groups to the imidazole group of L-histidine. Finally, the Trt protective group was shucked off to give histidine analogs under acidic and basic conditions, respectively.

The 1H NMR spectrum of L-his(R) is shown in Fig. 2. The main resonance signals were assigned to the corresponding protons, indicating successful synthesis.

Figure 1 Synthesis route of L-histidine analogs.

Figure 2 The 1H NMR spectrum of L-histidine analogs in CDCl3.

(A) 1H NMR(300 MHz, CDCl3): δ = 14.13(s,1H), 8.39(s,1H), 7.70(d,J = 7.5 Hz,2H), 7.48(d,J = 6.0 Hz,2H), 7.34(t,J = 7.2 Hz,2H), 7.26–7.21(m,1H), 7.03(s,1H), 6.25(d,J = 6 Hz,1H), 4.54(s,1H), 4.30–4.29(m,2H), 4.12–4.10(m,1H), 3.68(s,3H), 3.30–3.26(m,1H), 3.11–3.09(m,1H). (B) 1H NMR(300 MHz, CDCl3): δ = 12.88(s,1H), 8.54(s,1H), 7.72(d,J = 6.6Mz,3H), 7.52(d,J = 7.2 Hz,3H), 7.38(t,J = 7.5 Hz,3H), 6.42(d,J = 6.9 Hz,1H), 4.59(s,1H), 4.43–4.29(m,2H), 4.04–4.0(m,1H), 3.5–3.4(m,2H), 3.19(s,2H), 1.43–1.38(m,3H). (C) 1H NMR(300 MHz, CDCl3): δ = 12.77(s,1H), 8.55(s,1H), 7.72(d,J = 7.5 Hz,3H), 7.52(d,J = 6.3 Hz,3H), 7.35(t,J = 7.2 Hz,3H), 6.39(d,J = 5.7,1H), 4.61(s,1H), 4.46–4.29(m,3H), 4.20–4.11(m,1H), 3.26(s,2H), 1.44(s,6H). (D) 1H NMR(300MHz, CDCl3): δ = 8.45(s,1H), 7.71(d,J = 7.5 Hz,2H), 7.52(d,J = 7.2 Hz,2H), 7.35(t,J = 7.2 Hz,2H), 7.24–7.21(m,3H), 6.42(d,J = 5.7 Hz,1H), 4.59(s,1H), 4.35–4.29(m,2H), 4.12–3.95(m,3H), 3.20(s,2H), 1.66–1.64(m,2H), 1.23–1.21(m,2H), 0.86–0.82(m,3H).

The histidine of TH was substituted by these histidine analogs, and formed the TH analogs: methyl-TH, ethyl-TH, isopropyl-TH, and butyl-TH, respectively. The TH analog peptides were synthesized manually by the standard Fmoc-chemistry-based strategy.

Cellular uptake of TH analogs

To determine the pH-response of these new analogs and the role of alkyl moiety at the C-2 position of histidine, the cellular uptake of FITC-labeled TH analogs in HeLa cells was quantified by flow cytometry at various pH. As shown in Fig. 3, fluorescence intensity of FITC-methyl-TH at pH 6.5 and 6.0 was comparable to that at pH 7.4.

Figure 3 Flow cytometry analysis of Hela cells incubated with 5 µM FITC labeled TH analogs at different pH conditions (pH7.4, pH6.5 and pH6.0) for 1 h.

Hela cells incubated without FITC labeled TH analogues were used as control. (A) FITC-TH. (B) FITC-Methyl-TH. (C) FITC-Ethyl-TH. (D) FITC-Isopropyl-TH. (E) FITC-Butyl-TH. (F) Fluorescence intensity of internalized all peptides.

Interestingly, ethyl-TH exhibited lower fluorescence intensity than TH at pH 7.4 and 6.5, while fluorescence intensity of ethyl-TH at pH 6.0 increased substantially, to a quantitative level similar to that of TH at pH 6.0. It is shows that ethyl-TH increased its fluorescence intensity much greatly from pH 7.4 and 6.5 to pH 6.0 than TH. This suggested that the pH-responsive effect of ethyl-TH is better than TH.

Furthermore, the internalization ability of butyl-TH improved significantly when butylated histidine was introduced to the TH peptide. Fluorescence intensity of butyl-TH increased remarkably in comparison to that of TH at all pH values. However, butyl-TH presented approximately 1.72-fold higher fluorescence intensity than TH at pH 6.5, as well as higher fluorescence intensity than all the other TH analogues.

To evaluate the cellular uptake of new TH analogs at varying pH, confocal laser scanning microscopy (CLSM) was used. As shown in Fig. 4, methyl-TH displayed poor pH-responsiveness, since it showed lower cellular uptake than TH at both pH 6.5 and pH 6.0. In comparison with TH, ethyl-TH showed slightly higher fluorescence intensity at pH6.0relative to at pH7.4 and pH 6.5, indicating that ethyl-TH had a better pH-dependent cellular uptake than TH at pH 6.0. For isopropyl-TH, a small difference was found in the cellular uptake of isopropyl-TH between pH7.4 and pH6.5. The fluorescence intensity of isopropyl-TH at pH 6.5 is slightly lower than that at pH 7.4 in HeLa cells. Compared to TH, isopropyl-TH shows no pH-responsive at pH 6.5. When HeLa cells were treated with FITC-butyl-TH, significantly enhanced intensity was observed at all pH conditions. Compared to TH, butyl-TH was more pH-responsive at pH 6.5. The result was consistent with that obtained by flow cytometry analysis.

Figure 4 CLSM images of Hela cells incubated with 5 µM FITC labeled TH analogs at different pH conditions (pH7.4, pH6.5 and pH6.0) for 1 h.

(A) FITC-TH. (B) FITC-Methyl-TH. (C) FITC-Ethyl-TH. (D) FITC-Isopropyl-TH. (E) FITC-Butyl-TH.

Figure 5 Cytotoxicity of the TH analogs against the Hela cells for 2 h at different pH conditions.

Cytotoxicity was determined by the MTT assay. Data are the mean ± SD. ∗ indicates p < 0.05, ∗∗ indicates p < 0.01. (A) pH7.4. (B) pH6.5. (C) pH6.0.

In vitro cytotoxicity study

The cytotoxicity of TH analogs at different pH was evaluated with the MTT assay. As shown in Fig. 5A, none of the peptides have shown cytotoxicity against HeLa cells at pH7.4, with cell viability greater than 83% at the concentration of 20 µM. It is worth noting that methyl-TH didn’t show any significant cytotoxicity compared to the other three TH analogs at low pH (pH 6.5 and pH 6.0); this can probably be attributed to a decreased cell-penetrating activity of methyl-TH after the introduction of a methyl moiety (Fig. 3). This is consistent with a previous report (Song et al., 2011) that found that TP10 analogs exhibited lower penetrating activity with less cytotoxicity. It has also been observed that isopropyl-TH and butyl-TH didn’t show obvious cytotoxic variation from pH 7.4 to pH 6.0, indicating that the cytotoxicity of TH after modification with isopropyl and butyl did not increase proportionally with the increase in the length of alkylated groups. Curiously, the viability of cells treated with TH and ethyl-TH at different concentrations was higher than 80% at pH 7.4 and pH 6.5, indicating no significant cytotoxicity. However, when the pH value reached 6.0, the cytotoxicity of TH, and especially ethyl-TH, increased at the concentration of 20 µM.

LDH is always used as an indicator of acute membrane disturbance (Schinwald et al., 2012). In order to get a general view of the toxic profile displayed by TH analogs, the levels of LDH produced in the culture medium were measured at different pH condition. As shown in Fig. 6, the results of the LDH leakage assay were in line with the results of the MTT assay. At pH 7.4, TH and all the modified analogs did not cause a significant release of LDH. The LDH levels induced by methyl-TH, isopropyl-TH and butyl-TH at pH 6.5 and pH 6.0 were comparable to that at pH 7.4. LDH levels of TH and ethyl-TH rose as the pH value decreased. When the pH value dropped to 6.0, ethyl-TH showed the most obvious LDH release in comparison with TH and the other analogs.

Hemolysis assay for TH analogues

To evaluate the safety of TH analogs in the blood, a hemolysis assay was performed. As shown in Fig. 7, TH showed little cytotoxicity against red blood cells and TH analogs also showed remarkably low cytotoxicity in comparison with TH at the maximum concentration of 200 µM.

In vitro antitumor efficacy

The corresponding analogs were designed to improve the pH-responsive ability of TH. In order to further elucidate the introduced histidine analogs endue TH with the better pH-responsive capacity at acid tumor microenvironment for elevated cellular uptake and antitumor activity, the new TH analogs were attached to the 20-OH position of CPT by using a disulfide releasable carbonate linker, and the antiproliferation of these CPP-CPT conjugates were evaluated at different pH conditions in Hela cells. As shown in Fig. 8, there was no change in the cytotoxicity and selectivity of isopropyl-TH-CPT in comparison to TH-CPT at all investigated pH values. In contrast, after 72 h incubation, the antiproliferative effects of ethyl-TH-CPT and butyl-TH-CPT in HeLa cells were significantly different at pH 7.4, 6.5 and 6.0. Butyl-TH-CPT in particular showed a remarkably enhanced pH-responsive cytotoxicity at higher concentrations (10 µM and 20 µM). Moreover, although the cellular uptake level of ethyl-TH was lower than that of TH under the same pH (Fig. 2), the cytotoxicity of ethyl-TH-CPT at a concentration of 1.25 µM was approximately 3.44- and 3.83-fold higher than that of TH-CPT at pH 6.5 and pH 6.0, respectively. These results demonstrated that, compared with TH-CPT, the cytotoxicity and selectivity of ethyl-TH-CPT and butyl-TH-CPT to cancer cells can be greatly improved. As described above, the pH-responsiveness of the conjugate obtained after histidine isopropylation did not show a noticeable improvement in the pH range from 7.4 to 6.0. This data was consistent with earlier results demonstrating pH-dependent cellular uptake of TH analogs.

Figure 6 LDH leakage in Hela cells treated with TH analogs for 2 h at different pH conditions.

(A) pH7.4. (B) pH6.5. (C) pH6.0. (n = 3, mean ± SD). ∗ indicates p < 0.05, ∗∗ indicates p < 0.01.

Figure 7 Hemolytic activity of TH analogs on mice red blood cells.

(n = 3, mean ± SD). ∗ indicates p < 0.05, ∗∗ indicates p < 0.01.

Figure 8 Cytotoxicity of free CPT, TH-CPT and TH analogs-CPT toward Hela cells at different pH conditions after incubation for 30 min and further incubated for 72 h at various concentrations.

(A) CPT (B) TH-CPT (C) Ethyl-TH-CPT (D) Isopropyl-TH-CPT (E) Butyl-TH-CPT. (n = 3, mean ± SD). ∗ indicates p < 0.05, ∗∗ indicates p < 0.01.

Discussion

Generally, cationic CPPs can bind on the anionic membrane surfaces via electrostatic interaction and promote cellular uptake (Poon & Gariepy, 2007). Thus, cationic functionalityis a critical factor for cell penetration. Histidine is especially pH-responsive (Zaro, Fei & Shen, 2012; Ouahab et al., 2014). Due to the presence of an imidazole ring, it has the ability to receive protons and protonate into a positive charge in an acidic tumor environment, whereas it remains predominantly uncharged under normal physiological conditions. The pH-responsiveness of histidine has been widely utilizedin the design of pH-responsive CPPs for targeted drug delivery. The protonation ability of histidine’simidazole moiety is the key for pH-dependent cellular uptake of histidine-rich peptides. Thus, enhancing the degree of protonation of histidine in histidine-rich peptides will significantly increase their sensitivity to acidic conditions. In this regard, this study focused on the development of new histidine analogs that enhance the protonation of histidine by introducing alkylated groups with different electron-donating ability. Finally, after the introduction of such modified histidine analogs into TH, TH can get protonated more easily and exhibit a higher sensitivity in an acidic environment. In our study, different alkylated groups exhibited varying influence on the pH-response of TH. We found that introducing a methyl group did not improve the pH-responsiveness of TH. Methyl-TH was less hydrophobic than TH, which was confirmed by measuring its retention time on the C18 reverse-phase HPLC column (Table 1) (Song et al., 2011).The hydrophobicity of CPPs has been shown to facilitate their membrane translocation (Milletti, 2012; Wada et al., 2013). Clearly, a methyl moiety resulted in a less hydrophobic methyl-TH and in reduction of cellular uptake at all the pH values.

To determine the pH-response of these new analogs and the role of alkyl moiety at the C-2 position of histidine, the cellular uptake of FITC-labeled TH analogs in HeLa cells was quantified by flow cytometry at various pH. The cellular uptake level of ethyl-TH was lower than that of TH under the pH 7.4 and 6.5. However, the fluorescence intensity of ethyl-TH at pH 6.0 increased substantially, to a quantitative level similar to that of TH at pH 6.0. It is shows that ethyl-TH increased its fluorescence intensity much greatly from pH 7.4 and 6.5 to pH 6.0 than TH. This suggested that the pH-responsive effect of ethyl-TH is better than that of TH at lower pH values. Also, the cytotoxicity of ethyl-TH-CPT at a concentration of 1.25 µM was approximately 3.44- and 3.83- fold higher than that of TH-CPT at pH 6.5 and pH 6.0, respectively. These results demonstrated that compared to TH, ethyl-TH and butyl-TH exhibited an improved pH-responsive cellular uptake at pH 6.0 and pH 6.5, respectively. The degree of protonation of histidine at different pH values was directly correlated with the electron-donating ability of introduced alkylated groups. The longer carbon chain confers upon the alkylated groups an enhanced electron-donating ability, which results in varying histidine protonation at different pH values and a different pH-response in acidic environments. In addition, the results shows that the change of fluorescence discrepancy of isopropyl-TH from PH7.4 to PH6.0 was lower than that of TH , so isopropyl-modified TH shows a lower pH-dependent cellular uptake effective, indicating that the pH-response of isopropyl-TH is worse than that of ethyl-TH and butyl-TH. The superior acid-responsive effect of ethyl and butyl conjugates, compared with isopropyl conjugates, implies that straight-chain alkyl groups are better in modulating the degree of protonation under acidic conditions in comparison with branched alkyl groups. Accordingly, the alkylated conjugate of TH is a good representative of novel TH analogs with optimal pH-responsive and cellular uptake.

The cytotoxicity of CPPs is positively related to the degree of their positive charge (Jiang et al., 2012; Zhang et al., 2011; Song et al., 2011). This study focused on studying the in vitro cytotoxicity of these TH analogs. Our results suggested that the pH-responsiveness of ethyl-TH at pH 6.0 is better than that of TH and all the other TH analogs. The possible reason for this might be that easier protonation of ethyl-TH can increase the intensity of positive charge at pH 6.0, thereby resulting in higher cytotoxicity of ethyl-TH at higher concentrations. Accordingly, after alkylation, TH still exhibited slight cytotoxicity at physiological pH.

The safety of the drug vector is a vital aspect in any pharmaceutical application. In this study, we also performed a hemolysis assay for all TH analogues. The results of the assay suggested improved relative safety of TH analogs. It was reported that cationic CPPs may cause cell lysis and systemic toxicity (Jiang et al., 2012; Zorko & Langel, 2005) when used as delivery vectors, which restricts their successful in vivo utilization. Due to the loss of cationic charge at physiological pH, histidine-containing TH exhibited little cytotoxicity to normal cells, which is in agreement with previous literature (Zhang et al., 2011). Interestingly, after histidinealkylation, the new TH analogs exhibited relatively lower cytotoxicity at higher concentrations, suggesting a smaller degree of perturbation with the negatively charged membrane of erythrocytes. In comparison, the diminished cytotoxicity of new TH analogs to normal blood cells implied their safety as vectors for targeted drug delivery.

In general, the principal drawback of clinical conventional anticancer agents is their lack of specificity, leading to undesirable side effects. It has been reported that the acidic tumor microenvironment is one of the major obstacles for efficient drug delivery, and that there is a need for pH-responsive vectors for anticancer drugs (Shen et al., 2012). The pH-sensitive, cell penetrating ability of TH makes it suitable for targeted drug delivery to tumor cells with an acidic extracellular environment. The results of our study strongly demonstrated that, compared with TH-CPT, the cytotoxicity and selectivity of ethyl-TH-CPT and butyl-TH-CPT is greatly improved. Meanwhile, the pH-responsiveness of isopropyl histidine conjugates, did not show a noticeable improvement, but was consistent with the earlier results demonstrating a pH-dependent cellular uptake of TH analogs. In comparison with isopropyl-TH-CPT, ethyl-TH-CPT and butyl-TH-CPT exhibited a desirable pH-sensitivity to tumor cells. Therefore, after histidine modification, the improved pH-responsiveness of TH makes it suitable as a vector for targeted CPT delivery to tumor tissues with an acidic tumor microenvironment.

Conclusion

In this paper, by introducing electron donating groups to the histidine imidazole ring of TH, we developed a new type of acid-responsive CPP with a high sensitivity to the acidic tumor microenvironment and low toxicity under physiological conditions. This work opens a new avenue for designing preeminent pH-responsive CPPs, which could easily translocate into cells under weakly acidic conditions. The new TH analogs with modified histidine could easily protonate into a positive charge in a weakly acid microenvironment due to the introduction of electron donating groups. As a result, these TH analogs could deliver more CPT to the tumor cells and show excellent antitumor effects in an acidic extracellular environment, in comparison with TH-CPT and free CPT. These properties render the TH analogs we designed and synthesized suitable and superior as pH-sensitive vectors for targeted antitumor drug delivery with low toxicity. In view of this, TH analogs provided us with a new perspective on the development of pH-sensitive, CPP-modified drug delivery systems.

Supplemental Information

Supplemental Information 1 LDH-TH_pH_6.0

Click here for additional data file.

Supplemental Information 2 LDH-TH_pH_6.5

Click here for additional data file.

Supplemental Information 3 LDH-TH_pH_7.4

Click here for additional data file.

Supplemental Information 4 MTT-TH_pH_6.0

Click here for additional data file.

Supplemental Information 5 MTT-TH_pH_6.5

Click here for additional data file.

Supplemental Information 6 MTT-TH_pH_7.4

Click here for additional data file.

Supplemental Information 7 TH-Cellular_uptake

Click here for additional data file.

Supplemental Information 8 TH-Hemolysis_assay

Click here for additional data file.

Supplemental Information 9 TH-In_vitro_antitumor_efficacy

Click here for additional data file.

Additional Information and Declarations

Competing Interests

Author Contributions

Data Availability

The authors declare there are no competing interests.

Jia Yao conceived and designed the experiments, wrote the paper, reviewed drafts of the paper.

Yinyun Ma wrote the paper, reviewed drafts of the paper.

Wei Zhang and Li Li performed the experiments, prepared figures and/or tables.

Yun Zhang and Li Zhang performed the experiments.

Hui Liu analyzed the data.

Jingman Ni contributed reagents/materials/analysis tools, reviewed drafts of the paper.

Rui Wang contributed reagents/materials/analysis tools.

The following information was supplied regarding data availability:

The raw data has been supplied as a Supplementary File.

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
