# Peer review of "Design of new acid-activated cell-penetrating peptides for tumor drug delivery"

_PeerJ, doi:10.7717/peerj.3429_

## Round 0.1 · original submission · Major Revisions

· Academic Editor

Major Revisions

Please revise your manuscript accordingly.

·

Basic reporting

The manuscript was written with good clarity.

Sufficient background information was provided and relevant literatures cited.

The article structures are fine except that the raw data was provided in an unknown format and can not be opened.

The manuscript is "self-contained"

Experimental design

The research question raised by the manuscript is sound and meaningful.

The experiments were well designed with proper controls and the investigation was performed with high technical standard.

Methods were provided in good details.

Validity of the findings

1.Statistcal analysis is missing for all quantitative figures. Error bars are needed to tell whether the differences between groups are statistical significant.

2.Figure 2, the authors did not mention how FITC was labeled anywhere in the manuscript.

3.line 306-310, the authros wrote " Interestingly, ethyl-TH exhibited lower fluorescence intensity than TH at pH 7.4 and 6.5, while fluorescence intensity of ethyl-TH at pH 6.0 increased substantially, to a quantitative level similar to that of TH at pH 6.0. This suggested that the pH-responsive effect of ethyl-TH is better than that of TH at lower pH values." However, according to figure 2F, the variations of fluoresence intensity from pH7.4 to pH6.0 between TH and ethy-TH are almost the same. Therefore, the conclusion drawn by the authors is not accurate.

4.Line 342, according to Figure 4, the concentration of the peptide should be 20 uM instead of 40 uM, please verify. Also in Figure 2 and Figure 3, please indicate the concentration of the peptides used in the experiment.

5.HPLC and Mass spec results of the CPT-TH analog conjugates should be included to demonstrate the successful conjugation and the purity of the product.

6.Line 392-395. The author wrote :"Moreover, although the cellular uptake level of ethyl-TH was lower than that of TH under the same pH (Figure 1), the cytotoxicity of ethyl-TH-CPT at a concentration of 1.25 μM was approximately 3.44- and 3.83- fold higher than that of TH-CPT at pH 6.5 and pH 6.0, respectively." Can the authors speculate why ethyl-TH-CPT cytotoxicity is higher than TH-CPT if the uptake is lower as in figure 1? After drug conjugation, is the relative uptake efficiency still in the same order??

7.Line 321-322, the authors wrote " Fluorescence intensity of isopropyl-TH was lower in
HeLa cells than that of TH at both pH 7.4 and pH 6.5." However, according to Figure 3, it is difficult to draw this conclusion without quantitation. As a matter of fact, the fluorescence intensity of isopropyl-TH is actually higher than that of TH at pH7.4.

8.Line 435-436, the authors wrote "compared to TH, isopropyl-modified TH shows a lower pH-dependent cellular uptake at all pH values...". According to Figure 2F, the conclusion is not valid at least for pH7.4 and 6.5.

Additional comments

I am curious to see that the cell penetrating abilities of the TH analogs often do not correlated with their cytotoxicities, LDH releaseing effects etc. For example, the ethy-TH has a relatively low cell penetrating ability yet excels others in cytotoxicity and LDH releasing. It would be helpful to the readers if the authors can give speculative comments on this in the disccusion section.

Reviewer 2 ·

Basic reporting

no comment

Experimental design

no comment

Validity of the findings

no comment

Additional comments

The MS designed and synthesized a series of new alkylated histidine substituents and illustrated the new TH analogs (ethyl-TH and butyl-TH) had an optimal pH-response in an acidic environment. Especially the new TH analogs exhibited relatively lower toxicity than TH. In addition, this work explores a new avenue for the development of improved acid-activated, acid-activated CPPs as potential anticancer drug vectors for efficient targeted drug delivery. The MS is basically well prepared. The preparations, basic reporting and experimental design have a rather high quality. There are some errors of English usage and grammar that need to be addressed in an editorial revision. My specific suggestions for improvement of the paper, all relatively minor in nature, are given below (Some further problems are indicated in the PDF). Even though I had many suggestions and comments, I recommended “minor revision” because all of these involve only relatively minor rewrites of the MS, not major reorganizations or reconsiderations of results. Thus, here only some major problems are briefly listed:

1. Re-check your MS and corrected minor errors in English usage or smoothing out awkward sentences.
2. Careful read instruction to authors to avoid losing small changes like deletion or insertion of spaces.
3. References format is disorderly and unsystematic, such as whether the journal abbreviation or full spelling.

Annotated reviews are not available for download in order to protect the identity of reviewers who chose to remain anonymous.

·

Basic reporting

The manuscript by Jia Yao et. al identified several TH analogs, especially ethyl-TH and buty-TH, with higher sensitivity and selectivity in acidic conditions, which is much more helpful for drug delivery in tumors with acidic microenvironment. The authors should edit the English language and condense the description in introduction and discussion . Here is the detailed suggestions:

1) line 220
The Hela cell line was used during in vitro study. The manuscript should claim the source of the cell and whether the cells were tested for mycoplasma contamination

2) The statistical results should be added in the figures, such as Figure 2F, 4, 5, 6 and 7.

3) Figure2
The manuscript didn't show which TH was used in Figure 2A, B, C, D and E. It is difficult to follow the description in the results (line 340-351). I suggest the authors label TH, methy-TH... in the Figures or add Figure 2A, 2B ... in the results, not simply Figure 2.

Similar concerns for Figure3, 4, 5 and 7.

Experimental design

4) Figure 7
It will be better to show the IC50 of ethyl-TH-CPT, butyl-TH-CPT and TH-CPT in different pH conditions, 7.4, 6.5 and 6.0, which can directly demonstrate the distinct antitumor efficacy.

Validity of the findings

The main finding that ethyl-TH and buty-TH are more sensitive and selective in acidic condition. The authors do supply many evidence to demonstrate that.

---

## Round 0.2 · accepted · Accept

· Academic Editor

Accept

Thank you for your contribution. As you can see, the prior reviewers are satisfied by your revisions.

·

Basic reporting

No comments

Experimental design

No comments

Validity of the findings

No comments

Additional comments

All my questions have been well addressed

·

Basic reporting

no comment

Experimental design

no comment

Validity of the findings

no comment

Additional comments

no comment